# Association between Inflammatory Bowel Disease and Subsequent Development of Restless Legs Syndrome and Parkinson’s Disease: A Retrospective Cohort Study of 35,988 Primary Care Patients in Germany

**DOI:** 10.3390/life13040897

**Published:** 2023-03-28

**Authors:** Sven H. Loosen, Kaneschka Yaqubi, Petra May, Marcel Konrad, Celina Gollop, Tom Luedde, Karel Kostev, Christoph Roderburg

**Affiliations:** 1Department of Gastroenterology, Hepatology and Infectious Diseases, University Hospital Düsseldorf, Medical Faculty of Heinrich Heine University Düsseldorf, 40225 Duesseldorf, Germany; kaneschka.yaqubi@med.uni-duesseldorf.de (K.Y.);; 2FOM University of Applied Sciences for Economics and Management, 60549 Frankfurt am Main, Germany; 3Epidemiology, IQVIA, 60549 Frankfurt, Germany

**Keywords:** IBD, ulcerative colitis, UC, Crohn’s disease, CD, RLS

## Abstract

Background: In addition to the gastrointestinal symptoms, inflammatory bowel disease (IBD), which includes Crohn’s disease (CD) and ulcerative colitis (UC), is associated with extraintestinal manifestations, including neurological disorders, which are gaining increasing attention due to a recently increased focus on the gut–brain axis. Here we aim to evaluate the association between IBD and restless legs syndrome (RLS) as well as Parkinson’s disease (PD) in a cohort of primary care patients in Germany. Methods: The study included 17,994 individuals with a diagnosis of IBD (7544 with CD and 10,450 with UC) and 17,994 propensity-score-matched individuals without IBD from the Disease Analyzer database (IQVIA). An initial diagnosis of RLS or PD was assessed as a function of IBD. Associations between CD and UC with RLS and PD were analyzed using Cox regression models. Results: During the 10-year observation period, 3.6% of CD patients vs. 1.9% of matched non-IBD pairs (*p* < 0.001) and 3.2% of UC patients vs. 2.7% of matched pairs (*p* < 0.001) were diagnosed with RLS. The results were confirmed by Cox regression analysis, which showed a significant association between UC (HR: 1.26; 95% CI: 1.02–1.55) and CD (HR: 1.60; 95% CI: 1.23–2.09) and subsequent RLS. The incidence of PD in IBD patients was not significantly increased. However, we observed a non-significant trend towards a higher incidence of PD in male patients with CD but not UC (HR: 1.55; 95%CI: 0.98–2.45, *p* = 0.064). Conclusions: The present analysis suggests a significant association between IBD and the subsequent development of RLS. These findings should stimulate further pathophysiological research and may eventually lead to specific screening measures in patients with IBD.

## 1. Introduction

Inflammatory bowel disease (IBD) is a chronic, immune-mediated systemic condition commonly comprising Crohn’s disease (CD) and ulcerative colitis (UC). While symptoms of the gastrointestinal tract are at its core, IBD is also associated with extraintestinal manifestations and an increased risk for several co-morbidities [1]. In recent years, there has been growing concern about a heightened risk for the development of neurodegenerative and neuropsychiatric disorders in patients with IBD. The pathogenesis of these conditions is far from understood, but several mechanisms have been proposed, including neuroinflammation, alterations to the gut–brain axis, gastrointestinal dysbiosis, increased intestinal permeability, and nutrient deficiency and other metabolic changes [2,3].

Interestingly, some epidemiological studies have produced conflicting results. In the case of neurological conditions characterized by dopaminergic dysfunction, such as Parkinson’s disease (PD) and restless legs syndrome (RLS), some studies have suggested a higher prevalence of both PD [4,5,6] and RLS [7,8] in patients with IBD compared with the general population. At the same time, conflicting results have recently been observed [9], and the underlying pathophysiology between IBD and these neurological conditions remains highly controversial [10]. Importantly, several studies have provided robust evidence that both RLS and PD are associated with a significantly reduced quality of life [11,12,13,14], which justifies more research for a better understanding and identification of the risk factors for these conditions in order to provide prevention strategies and adequate therapy for affected individuals.

In the present manuscript, we aimed to assess a possible association between IBD and the subsequent occurrence of two distinct neurological conditions—PD and RLS—in a large cohort of outpatients in Germany. By exploring this relationship, we hope to contribute to a greater understanding of the complex interplay between the gut and the brain and the possible pathophysiological relationship in the context of these conditions. Such data may lead the way to improvements in the clinical management and therapy of these conditions, providing a potential impact on patient health.

## 2. Materials and Methods

### 2.1. Database

This retrospective cohort study was based on electronic medical records (EMR) from the Disease Analyzer (DA) database (IQVIA). The DA database, which has already been used in several previous studies focusing on IBD or PD, contains anonymous EMR data on diagnoses, prescriptions, as well as basic medical and demographic data from office-based practices [15]. The database covers approximately 3–5% of all office-based practices in Germany. The sampling method for the DA database uses statistics from the German Medical Association to determine the panel design according to specialist group, German federal state, community size category, and age of physician. It has previously been shown that the panel of practices included in the Disease Analyzer database is representative of general and specialized practices in Germany [15].

### 2.2. Study Population

The present study included patients aged ≥40 years with a diagnosis of IBD including Crohn’s disease (ICD-10: K50) and ulcerative colitis (ICD-10: K51) from 1284 general practices in Germany between January 2005 and December 2020 (index date; Figure 1). The first documented diagnosis of IBD within this time period was considered the index date. Further inclusion criteria were an observation time of at least 12 months prior to the index date and a follow-up time of at least 6 months after the index date. On average, study patients were followed up for 6.2 years (minimum 6 months, maximum 10 years). Patients with pre-existing diagnoses of extrapyramidal and movement disorder (ICD-10: G20–G26) prior to or at index date were excluded. After applying the same inclusion criteria, individuals without IBD were matched to IBD patients using propensity score matching (1:1) based on age (±2 years), sex, average yearly consultation frequency during the follow-up, diabetes (ICD-10: E10–E14), head injury (ICD-10: S00–S09), and depression (ICD-10: F32, F33), diagnoses documented within 12 months prior to the index date. For the non-IBD cohort, the index date was defined as a randomly selected visit between January 2005 and December 2020 (Figure 1). Diabetes head injury, and depression were included as these diagnoses are known to be associated with PD or RLS.

### 2.3. Study Outcomes and Statistical Analyses

The outcomes of the study were an initial diagnosis of PD (ICD-10: G20) or RLS (ICD-10: G25.8 + original diagnosis text) during the subsequent up to 10 years following the index date as function of IBD. Differences in sample characteristics and diagnosis prevalence between IBD and non-IBD cohorts were compared using the Wilcoxon signed-rank test for continuous variables, the McNemar test for categorical variables with two categories, and the Stuart–Maxwell test for categorical variables with more than two categories. The 10-year cumulative incidence of PD and RLS in the cohort with and without IBD was further assessed by means of Kaplan–Meier (KM) analysis, and the KM curves were compared using the log-rank test. Finally, an univariable Cox regression analysis was conducted to assess the association between IBD and PD, as well as between IBD and RLS. Associations with CD and UC were analyzed separately. Additionally, Cox regression analyses were conducted separately for men and women. Results of the Cox regression model are displayed as hazard ratios (HR) and 95% confidence intervals (CI). A *p*-value of <0.05 was considered statistically significant. Analyses were carried out using SAS version 9.4 (SAS Institute, Cary, NC, USA).

## 3. Results

### 3.1. Basic Characteristics of the Study Sample

The present study included 17,994 individuals with a diagnosis of IBD (7544 with CD and 10,450 with UC) as well as 17,994 matched individuals without IBD. The basic characteristics of study patients are shown in Table 1. The mean patient age was 59.7 (standard deviation (SD): 12.8) years, and 55.5% of patients were women. On average, patients visited their GPs 10.7 times per year during the follow-up. Due to the matched pairs design, there were no significant differences between both cohorts in terms of age, sex, visit frequency, and predefined co-diagnoses (Table 1).

### 3.2. Association of IBD and A Subsequent Diagnosis of Restless Legs Syndrome

Within the 10-year observation period, 3.6% of CD patients vs. 1.9% of matched non-IBD patients (*p* < 0.001), as well as 3.2% of UC patients vs. 2.7% of matched patients (*p* < 0.001) were diagnosed with RLS (Figure 2). The incidence of RLS was higher in patients with CD than in patients without IBD (3.5 cases per 1000 patient years in the CD cohort and 2.2 cases per 1000 patient years in the matched non-IBD cohort). Similarly, a higher proportion of patients with UC compared to those without UC developed RLS (the incidence of RLS per 1000 patient years in the UC cohort was 3.5 cases and 2.7 cases in the matched non-IBD cohort). There was a significant positive association between CD and subsequent RLS (HR: 1.60; 95% CI: 1.23–2.09), which was confirmed for both women (HR: 1.58; 95% CI: 1.14–2.20) and men (HR: 1.64; 95% CI: 1.04–2.61, Table 2). Likewise, a significant positive association between UC and RLS was found for the total population (HR: 1.26; 95% CI: 1.02–1.55), and confirmed in women (HR: 1.49; 95% CI: 1.02–2.18) but not in men (HR: 1.18; 95% CI: 0.91–1.51, Table 2).

### 3.3. Association of IBD and A Subsequent Diagnosis of Parkinson’s Disease

During the 10-year follow-up period, 2.1% of CD patients vs. 1.8% of matched non-IBD pairs (*p* = 0.199, Figure 3), as well as 2.1% of UC vs. 2.1% of matched pairs (*p* = 0.757, Figure 3) were diagnosed with PD. The incidence was 2.1 cases per 1000 patient years in the CD cohort vs. 1.7 cases per 1000 patient years in the matched non-IBD cohort, and 2.1 cases per 1000 patient years in the UC cohort vs. 2.2 cases per 1000 patient years in the matched non-IBD cohort. We found no significant association between CD or UC and subsequent PD in the total study cohort (HR_CD_: 1.23; 95%CI: 0.90–1.69 and HR_UC_: 0.96; 95%CI: 0.75–1.23). However, we observed a trend towards a higher frequency of PD among male patients with CD but not UC (HR: 1.55; 95%CI: 0.98–2.45, *p* = 0.064).

## 4. Discussion

In patients with inflammatory bowel disease, a plethora of different co-morbidities have been described, including cardiovascular, metabolic, and neuropsychiatric conditions. These conditions can substantially increase the burden of disease and are associated with worse quality of life [1]. In the setting of neuropsychiatric and neurodegenerative diseases, an interplay between the gut, immune system, and the nervous system—often referred to as the gut–(immune)–brain axis—has been postulated as a possible link between diseases of the central nervous system (CNS) and IBD [2].

In the present study, we focused on analyzing a possible association between the occurrence of IBD and the subsequent diagnosis of RLS and/or PD. RLS and PD are common neurological disorders for which a certain overlap in pathophysiology has been proposed. For both conditions, it was demonstrated that the dopaminergic system and dopamine dysfunction play a vital role in disease etiology and therapy [16]. Anatomically, alterations in the nigrostriatal dopaminergic system seem to link both conditions [17,18]. In addition, both RLS and PD exhibit a clinical response to dopaminergic agents [16,19]. Despite these similarities, clinical association studies have led to mixed results. While some studies found a higher prevalence of RLS in PD patients compared to control subjects [20,21], other studies failed to provide evidence for an increased co-morbid occurrence of RLS and PD [22,23]. These differences might be elucidated when comparing the role of iron metabolism in both conditions. In PD (or other neurodegenerative diseases such as Alzheimer’s disease or amyotrophic lateral sclerosis), a dysregulation of iron metabolism has been proposed as a possible mechanism for neuronal cell death [24]. In contrast, RLS is associated with iron deficiency and with a reduction in iron concentrations in certain areas of the brain [25,26].

In the present study, we were able to show a significant correlation between IBD and RLS in a cohort of primary care patients in Germany, whereas no such correlation was found for IBD and PD. While the positive association of IBD and subsequent RLS was confirmed in the total population for CD, in patients with UC, an association for subsequent RLS was only found to be significant in women. These results are generally on a par with findings from previous studies analyzing associations between IBD and RLS [7,27]. Considering the role of iron concentrations in the pathophysiology of RLS, this association could be partially due to the coherence of chronic inflammation of the bowel and iron deficiency. Both CD and UC are inflammatory conditions that are not only restricted to the gastrointestinal tract but are also accompanied by systemic inflammation. This chronic and systemic inflammation is thought to contribute to the development of extraintestinal manifestations [1,28]. Sustained conditions of chronic inflammation are commonly linked to states of iron deficiency and associated diseases such as anemia of chronic disease [29,30]. Mechanistically, levels of the hepatic peptide hormone hepcidin are crucial in controlling iron metabolism by regulating serum iron concentrations, intestinal iron absorption, and intracellular iron sequestration [30]. During inflammation, the production and release of hepcidin is induced, thereby reducing iron availability. This process is controlled by a number of different pro-inflammatory cytokines, in particular by interleukin-6 [29,30]. In IBD, interleukin-6 is believed to be critically involved in its pathogenesis and in fueling the maintenance of chronic inflammation [31,32]. As well as the loss of iron due to intestinal bleeding, this state of systemic and chronic inflammation is believed to lead to iron deficiency and, for example, subsequent anemia in IBD [30,33]. Prolonged iron deficiency could therefore result in iron deficiency in the CNS, thereby contributing to the increased prevalence of RLS in patients with IBD. In addition, this mechanism could provide a possible explanation for the sex difference we observed in subsequent RLS in patients with UC since women are at higher risk for developing iron deficiency compared to men [34,35]. Further studies are required to assess iron deficiency as a possible mediator in the risk of developing RLS in patients with IBD.

In the case of PD, several epidemiological studies including PD patients from Taiwan [36], the United States [37], and Sweden [4] were able to provide evidence for a potential link between PD and IBD. In contrast, a study examining Medicare beneficiaries showed an inverse association between PD and IBD [38]. In the past, several mechanisms have been proposed for the observed evidence connecting PD and gastrointestinal dysfunction. One of the first studies hypothesizing a potential link between the gastrointestinal system and PD was provided by Braak et al. [39]. Braak and colleagues hypothesized that a potential pathogen would induce α-synucleinopathy in the periphery (e.g., through the nasal cavity and the olfactory system, or the enteric nervous system), which then ascends into the CNS through retrograde transport [39,40,41]. This hypothesis is supported by studies demonstrating pathological α-synuclein aggregates in the enteric nervous system and in gastrointestinal biopsies from PD patients [42,43]. Indeed, gastrointestinal inoculation of α-synuclein in mice was found to result in α-synuclein pathology in the brainstem similar to that observed in early-stage PD. Inoculation with α-synuclein in this mouse model was also found to lead to a reduction in dopaminergic neurons, and caused motor and non-motor symptoms characteristic for PD. Mice undergoing vagotomy prior to the inoculation event showed no PD-characteristic changes, suggesting a retrograde mechanism through the enteric nervous system and via the vagus nerve [44,45]. In IBD, increased gastrointestinal inflammation might trigger or accelerate this process, leading to abnormal α-synuclein deposition in the CNS [46,47].

In addition, gastrointestinal inflammation is thought to influence PD pathogenesis and progression due to overlapping inflammatory pathways [47,48]. Pro-inflammatory cytokines, such as IL-1β, TNF, IFNγ, IL-2, IL-6, and CXCL8, have been found to be increased in concentration in the cerebrospinal fluid and brain tissue of patients with PD [49,50]. Many of these cytokines are also strongly involved in the pathogenesis and pathophysiology of IBD. It is believed that long-term peripheral and gastrointestinal inflammation might promote inflammatory processes in the CNS, thereby contributing to neurodegenerative diseases such as PD [47,48]. In particular, TNF (tumor necrosis factor) seems to be a major cytokine involved in these processes [49,50]. In PD, mRNA expression of different pro-inflammatory cytokines, including TNF, was shown to be significantly increased in colon biopsies [51]. This is further supported by studies indicating that patients receiving anti-TNF agents are less likely to develop PD, reducing their risk by up to 92% in the case of PD and CD [37,52]. Notably, anti-inflammatory treatment with mesalazine or sulfasalazine in patients with IBD was also associated with a decreased risk of co-morbid PD [53,54], further highlighting the potential link between chronic gastrointestinal inflammation and neurodegenerative disease in the CNS.

Our data suggest that there may not be a significant link between these two conditions in this German primary care cohort, leaving different causes up for debate. For instance, differences in application rates of immunosuppressant medications—in particular, anti-TNF agents—might act on the observed conflicting results. Based on the experimental association between chronic peripheral inflammation and neuroinflammation, a preceding diagnosis of IBD could potentially lower the risk of subsequent PD due to the early use of anti-inflammatory agents [38]. These differences in clinical association between PD and IBD shed light on the complexity of the relationship between PD and IBD and highlight the need for further research to fully understand the underlying mechanism of both diseases.

Our study has several limitations. First, we must assume that in an outpatient treatment setting, some diagnoses may be coded incorrectly or misclassified by the attending physician. Second, the Disease Analyzer database does not provide information on, for example, the socioeconomic status (e.g., education and income of patients) or lifestyle-related risk factors (e.g., smoking, alcohol consumption, and physical activity), which might lead to a respective bias. No data on diagnosis methods were available and diagnoses information were captured by the general practitioner (GP) in charge. It is possible that GPs document diagnoses that were previously made in a hospital setting or by a specialist and do not partake in the diagnostic process. Finally, based on the study design, only associations and no causal relationships can be assumed.

Together, our data suggest a significant association between IBD and the subsequent development of RLS in a large real-world cohort of outpatients in Germany. We believe that these findings should stimulate further pathophysiological research and may eventually lead to specific screening measures in patients with IBD.

## Figures and Tables

**Figure 1 life-13-00897-f001:**
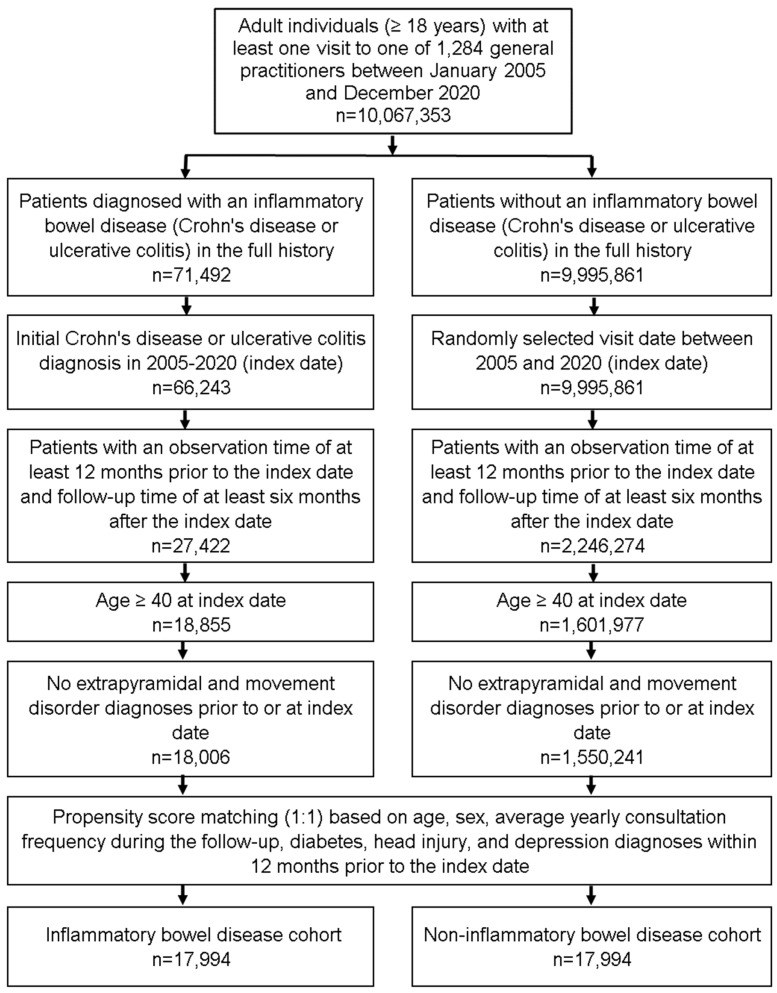
Selection of study patients.

**Figure 2 life-13-00897-f002:**
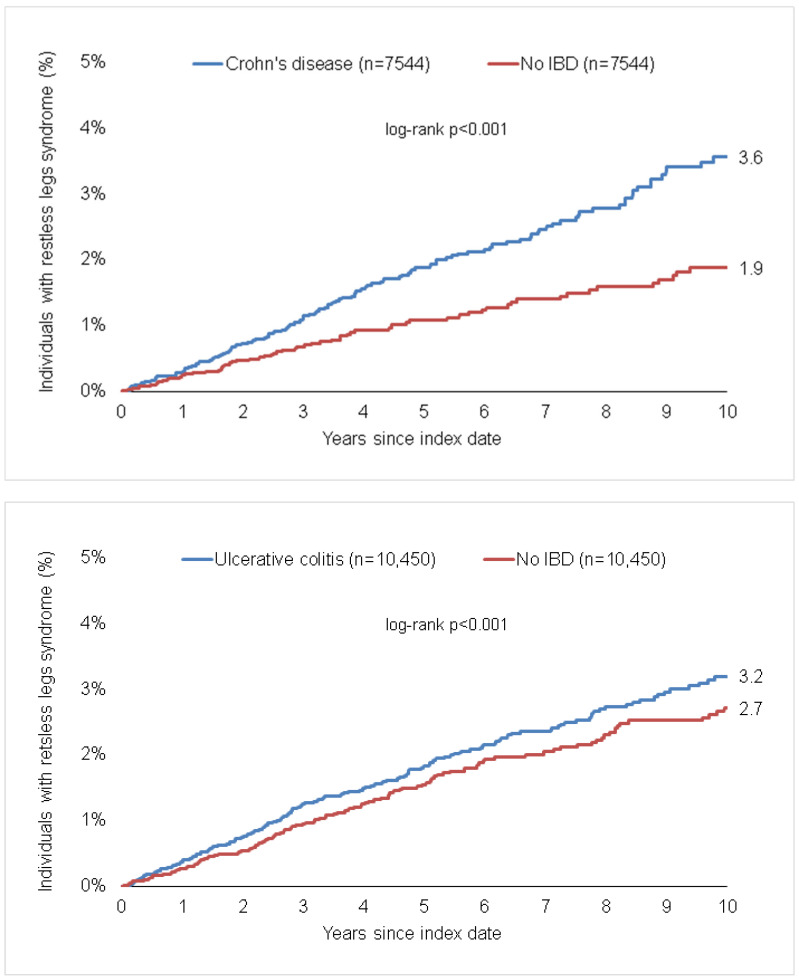
Cumulative incidence of restless legs syndrome in individuals with and without Crohn’s disease and ulcerative colitis.

**Figure 3 life-13-00897-f003:**
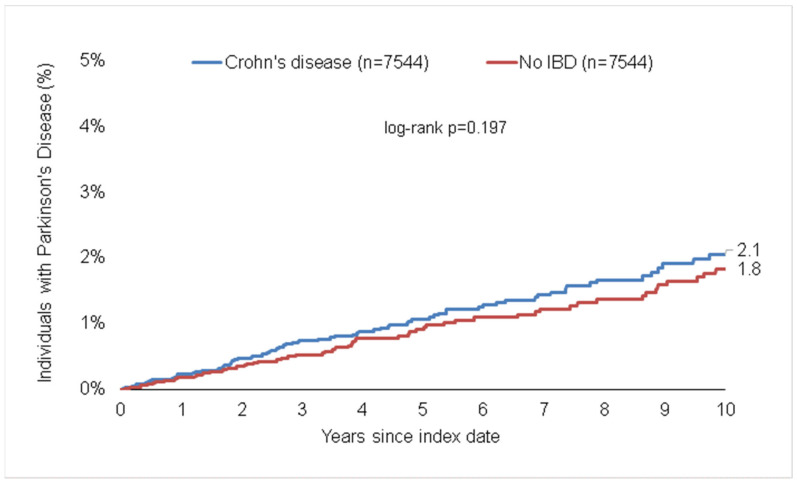
Cumulative incidence of Parkinson’s disease in individuals with and without Crohn’s disease and ulcerative colitis.

**Table 1 life-13-00897-t001:** Baseline characteristics of the study sample (after propensity score matching).

Variable	Proportion among Individuals with IBD (%) *n* = 17,994	Proportion among Individuals without IBD (%) *n* = 17,994	*p*-Value
Age (Mean, SD)	59.7 (12.8)	59.7 (12.8)	0.759
Age 40–50	29.4	29.1	0.535
Age 51–60	27.0	27.1	
Age 61–70	20.0	20.6	
Age 71–80	16.6	16.2	
Age > 80	7.0	7.0	
Women	55.5	55.5	0.958
Men	44.5	44.5	
Number of physician visits per year during the follow-up (Mean, SD)	10.7 (9.0)	10.6 (9.0)	0.101
Diabetes	22.7	22.6	0.940
Depression	24.8	24.8	0.971
Head injury	4.6	4.6	0.980

Proportions of patients given in %, unless otherwise indicated. SD: standard deviation.

**Table 2 life-13-00897-t002:** Association between Crohn’s disease and ulcerative colitis and subsequent Parkinson’s disease and restless legs syndrome (univariable Cox regression models).

Patient Group	Crohn’s Disease		Ulcerative Colitis	
	HR (95% CI)	*p*–value	HR (95% CI)	*p*–value
Parkinson’s Disease				
Total	1.23 (0.90–1.69)	0.198	0.96 (0.75–1.23)	0.759
Women	0.99 (0.63–1.54)	0.946	0.90 (0.63–1.28)	0.561
Men	1.55 (0.98–2.45)	0.064	1.02 (0.72–1.43)	0.927
Restless Legs Syndrome				
Total	1.60 (1.23–2.09)	<0.001	1.26 (1.02–1.55)	0.031
Women	1.58 (1.14–2.20)	0.006	1.18 (0.91–1.51)	0.210
Men	1.64 (1.04–2.61)	0.035	1.49 (1.02–2.18)	0.041

## Data Availability

The data is available upon reasonable request from the corresponding author.

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
