# Peer review of "Association between Inflammatory Bowel Disease and Subsequent Development of Restless Legs Syndrome and Parkinson’s Disease: A Retrospective Cohort Study of 35,988 Primary Care Patients in Germany"

_life, 2023, doi:10.3390/life13040897_

Round 1

Reviewer 1 Report

The manuscript is very well written and the topic is novel. Sufficient data has been provided to support the subject of the research. 

Author Response

We are grateful to the reviewer for his/ her positive feedback recommending our manuscript for immediate publication.

Reviewer 2 Report

Dear Authors,

The purpose of the manuscript was to assess a possible association between inflammatory bowel disease (IBD) and the subsequent occurrence of two distinct neurological conditions – Parkinson's disease (PD) and restless legs syndrome (RLS) – in a large cohort of outpatients in Germany. The study contributes to a greater understanding of the complex interplay between the gut and the brain and possible pathophysiological relationship in the context of these conditions. The retrospective cohort study was conducted on electronic medical records (EMR) from the Disease Analyzer (DA) database (IQVIA). The incidence of RLS and PD was calculated in IBD and non-IBD groups.

The following questions should be considered:

1.       What was the power of the study? The percentage or incidence was quite small to consider the significance.

2.       The discussion section should present similar research in this area. Please, keep the scope of the study when discussing the results obtained Give criticisms and limitations of the study. In my opinion, too much attention has been paid to iron metabolism and markers of inflammation.

My overall comment is that the article is not ready for publication, as it has it has flaws in methodology.

Author Response

The purpose of the manuscript was to assess a possible association between inflammatory bowel disease (IBD) and the subsequent occurrence of two distinct neurological conditions – Parkinson's disease (PD) and restless legs syndrome (RLS) – in a large cohort of outpatients in Germany. The study contributes to a greater understanding of the complex interplay between the gut and the brain and possible pathophysiological relationship in the context of these conditions. The retrospective cohort study was conducted on electronic medical records (EMR) from the Disease Analyzer (DA) database (IQVIA). The incidence of RLS and PD was calculated in IBD and non-IBD groups.

We appreciate the clear and constructive comments and suggestions of the reviewer. By providing some important clarifications we hope that the manuscript is now considered suitable for publication.

What was the power of the study? The percentage or incidence was quite small to consider the significance.

The power of the study was calculated for both CD and UC. To know the power, we need to know the differences in cumulative incidence of PD and RLS as well as absolute patient numbers. In the case of RLS the power is more than 0.90, and by PD it is 0.80. Not the power is the problem. By RLS we have in both CD and UC p-values of <0.001 showing that differences were strong enough and sample sizes large enough. In the case of PD we had very small differences between CD and non IBD and we had even no differences between UC and non-IBD. This was the reason why we did not achieve p-values <0.05. 

The discussion section should present similar research in this area. Please, keep the scope of the study when discussing the results obtained Give criticisms and limitations of the study. In my opinion, too much attention has been paid to iron metabolism and markers of inflammation.

We are grateful for this important comment regarding the discussion section of the manuscript. Based on this comment as well as a similar comment of reviewer#3, we have removed wide parts of the respective paragraphs from the new discussion section of the revised manuscript.

My overall comment is that the article is not ready for publication, as it has it has flaws in methodology.

We acknowledge the reviewer's concerns. In the revised manuscript we have addressed all comments and recommendations of this and the other reviewers to resolve all methodological issues raised. Moreover we want to highlight that throughout the manuscript, we used standard epidemiological  methodology including K-M- curves and Cox regression analyses, which are considered as best practice methods for the estimation of cumulative incidence.

Reviewer 3 Report

Dear Authors,

The manuscript "Association between Inflammatory Bowel Disease and the Subsequent Development of Restless Legs Syndrome and Parkinson's Disease - A Retrospective Cohort Study of 35,988 Primary Care Patients in Germany" is well, clearly, and understandably written.
It contains relevant information in the introductory section about the association between inflammatory bowel disease and restless leg syndrome and Parkinson's disease, which is supported by collected data.
After careful reading of the manuscript, some minor comments/questions arise that should be addressed before further consideration of its suitability for publication.
There are no lines in the pdf file, so I will have to phrase my suggestions this way:

·         Sentence - The incidence of RLS was 3.5 cases per 1,000 patient-years in the CD cohort and 2.2 cases per 1,000 patient-years in the matched non-IBD cohort. - should be written differently, it is not clear, one cannot understand what the authors wanted to say

·         Table 3 is missing, although it is mentioned in the text.

·         IMPORTANT
The "Discussion" section in the paper is based on assumptions that are not supported by any analysis in the text. Would it be possible for the authors to have at least some data on the biochemical status of the patients? Data on iron concentration would be particularly useful, since the discussion is largely based on these data. The paper would benefit from this information.

After a careful reading of the manuscript, I believe that these comments should be considered before further evaluating its suitability for publication.

Author Response

The manuscript "Association between Inflammatory Bowel Disease and the Subsequent Development of Restless Legs Syndrome and Parkinson's Disease - A Retrospective Cohort Study of 35,988 Primary Care Patients in Germany" is well, clearly, and understandably written. It contains relevant information in the introductory section about the association between inflammatory bowel disease and restless leg syndrome and Parkinson's disease, which is supported by collected data.

We are grateful to the reviewer for his/ her positive feedback recommending our manuscript for publication after minor revisions.

Sentence - The incidence of RLS was 3.5 cases per 1,000 patient-years in the CD cohort and 2.2 cases per 1,000 patient-years in the matched non-IBD cohort. - should be written differently, it is not clear, one cannot understand what the authors wanted to say

We have amended the sentence as following “The incidence of RLS was higher in patients with CD than in patients without IBD (3.5 cases per 1,000 patient years in the CD cohort and 2.2 cases per 1,000 patient-years in the matched non-IBD cohort). Similarly, a higher proportion of patients with UC compared to those without UC developed RLS (the incidence of RLS per 1,000 patient-years in the UC cohort was 3.5 cases and 2.7 cases in the matched non-IBD cohort).“ (page 4 of the revised manuscript).

Table 3 is missing, although it is mentioned in the text.

We are grateful to the reviewer for carefully reading the manuscript. Indeed, there is no table 3 and the respective term has been removed from the manuscript.

The "Discussion" section in the paper is based on assumptions that are not supported by any analysis in the text. Would it be possible for the authors to have at least some data on the biochemical status of the patients? Data on iron concentration would be particularly useful, since the discussion is largely based on these data. The paper would benefit from this information.

We are grateful for this important comment regarding the discussion section of the manuscript. Based on this comment as well as a similar comment of reviewer#2 suggesting that too much attention has been paid to iron metabolism and markers of inflammation we have removed wide parts of the respective paragraph from the new discussion section of the revised manuscript.

Round 2

Reviewer 2 Report

Dear Authors,

The manuscript has been improved. However, authors should carefully check the list of references, as references 31-38 are not mentioned in the text.

Author Response

We thank the reviewer for her/his careful evaluation and have corrected the mistake in the citations accordingly.